# Hematological abnormalities and associated factors among metabolic syndrome patients at the University of Gondar comprehensive specialized hospital, Northwest Ethiopia

**Amanuel Kelem**[1]*, **Elias Shiferaw**[2], **Tiruneh Adane**[2]

**1** Department of Medical Laboratory Sciences, Asrat Woldeyes Health Science Campus, Debre Berhan University, Debre Berhan, Ethiopia, **2** Department of Hematology and Immunohematology, School of Biomedical and Laboratory Sciences, College of Medicine and Health Sciences, University of Gondar, Gondar, Ethiopia

* amanuelkelem@gmail.com

**Data Availability Statement:** All relevant data are within the paper and its Supporting Information files.

## Abstract

### Background

Metabolic Syndrome (MetS) is a cluster of interconnected metabolic diseases. Hematological abnormalities are common but neglected complications of MetS. Thus, this study aimed to determine the magnitude of hematological abnormalities and their associated factors among MetS patients at the University of Gondar comprehensive specialized hospital, Northwest Ethiopia.

### Method

A hospital-based cross-sectional study was conducted at the University of Gondar comprehensive specialized hospital from March to May 2022. A total of 384 MetS patients were selected using a systematic random sampling technique. Data were collected using pre-tested structured questionnaires and checklists. Anthropometric and blood pressure measurements were taken, and blood sample was collected for complete blood count determination. Stool and blood film examinations were performed to detect intestinal and malaria parasites, respectively. Data were entered into EpiData 3.1 and analyzed by Stata 14.0 software. Bivariate and multivariate logistic regression models were fitted to identify factors associated with hematological abnormalities. A p-value of < 0.05 was considered statistically significant.

### Results

The magnitude of anemia, leukopenia, leukocytosis, thrombocytopenia, and thrombocytosis was found to be 13.3%, 0.5%, 2.9%, 1.6%, and 2.3%, respectively. Being male (AOR = 2.65, 95% CI: 1.14, 6.20), rural residency (AOR = 5.79, 95% CI: 1.72, 19.51), taking antihypertensive medications (AOR = 3.85, 95% CI: 1.16, 12.78), having elevated triglyceride

**Funding:** The study was funded by the University of Gondar. The funders had no role in study design, data collection and analysis, decision to publish, or preparation of the manuscript.

**Competing interests:** The authors have declared that no competing interests exist.

**Abbreviations:** BMI, Body Mass Index; BP, Blood Pressure; DBP, Diastolic Blood Pressure; DM, Diabetes Mellitus; FBS, Fasting Blood Sugar; Hb, Hemoglobin; HDL-C, High-Density Lipoprotein Cholesterol; IDF, International Diabetes Federation; IL, Interleukin; IR, Insulin Resistance; MCV, Mean Corpuscular Volume; MetS, Metabolic Syndrome; MPV, Mean Platelet Volume; PDW, Platelet Distribution Width; SBP, Systolic Blood Pressure; T2DM, Type 2 Diabetes Mellitus; TG, Triglyceride; TNF, Tumor Necrosis Factor; TPO, Thrombopoietin; WBC, White Blood Cell; WC, Waist Circumference; WHO, World Health Organization.

level (AOR = 2.21, 95% CI: 1.03, 4.75), and being overweight or obese (AOR = 0.32, 95% CI: 0.16, 0.64) were significantly associated with anemia.

## Conclusions

Anemia was the most prevalent hematological abnormality identified in the present study, followed by leukocytosis and thrombocytosis. Anemia was a mild public health problem among MetS patients in the study area. Routine anemia screening for all MetS patients, especially for those with significant associated factors, may help in the early detection and effective management of anemia, which subsequently improves the patients' quality of life.

## Introduction

Metabolic Syndrome (MetS) is a group of metabolic disorders that includes central obesity, insulin resistance (IR), hypertriglyceridemia, hypercholesterolemia, hypertension, and low high-density lipoprotein cholesterol (HDL-C) [1]. Various international organizations including the international diabetes federation (IDF) have proposed different criteria to define MetS [2–5]. According to IDF definition, for a person to be defined as having MetS, they must have central obesity along with any two of the following additional factors: raised triglyceride (TG) level, reduced HDL-C, raised blood pressure (BP), and raised fasting plasma glucose [4].

Although genetic susceptibility certainly plays a role, abdominal obesity is the major cause of MetS [6]. In addition, smoking, sedentary lifestyle, unhealthy diet, and lack of exercise all contribute to the development of MetS [7]. Available evidence showed that MetS is associated with a two-fold greater risk of cardiovascular disease and a five-fold increased risk of type 2 diabetes mellitus (T2DM) [8]. Additionally, people with MetS appear to be susceptible to several conditions such as asthma, fatty liver, sleep disturbances, cholesterol gallstones, polycystic ovary syndrome, and some types of cancer [3].

Metabolic syndrome is becoming a worldwide issue [9], and in recent years, both developed and developing countries have seen a sharp rise in the prevalence of the condition [10]. Currently, over a billion people worldwide suffer from MetS [7], with the global epidemic proportions of MetS to be around 20 to 25% [11]. An emerging high prevalence of MetS has been shown in Ethiopia, with the pooled prevalence being 34.9% [12] and an overall prevalence of 11.2% was found particularly in Gondar, Northwest Ethiopia, according to the national cholesterol education program adult treatment panel III criteria. However, compared to system resources devoted to other illnesses, the MetS epidemic and its care management has not received enough attention [13].

Hematological abnormalities, particularly anemia, are a highly prevalent complications among MetS patients, especially in those with T2DM and people who are overweight or obese [14–19]. The prevalence of anemia among MetS patients was reported to be 25.3% in Ethiopia [19] and a higher prevalence of anemia among T2DM patients was also reported in different parts of the world, ranging from 34.8% in Ethiopia to 66% in India [14,16,20,21]. Elevated Leukocytes and platelets were also reported to occur in MetS patients, even though the percentages were not clearly stated [22–24].

The mechanisms explaining the association between hematological abnormalities and MetS are not fully elucidated, but several possibilities have been proposed. Hematological markers are thought to be related to IR, which is regarded to be a key component of MetS [25]. Insulin resistance-induced vascular dysfunction may lead to a decreased production of prostacyclin

and nitric oxide by the vascular endothelium, which would promote an increase in platelet activity and a subsequent production of more platelets [24]. Endothelial dysfunction in MetS is also linked to elevated leukocyte count by enhancing leukocyte differentiation [26].

Additionally, chronic inflammation in MetS induces the synthesis of proteolytic enzymes and several cytokines such as interleukin (IL)-6 and tumor necrosis factor (TNF)-α. These might impair endothelial functioning and integrity, which could lead to an increase in leukocytes and change platelet count levels [24,27]. Inflammatory conditions are also known to cause an increase in thrombopoietin (TPO) resulting in an elevated platelet count [28]. In contrary, pro-inflammatory cytokines have been linked with the inhibition of erythroid progenitor cells, which leads to anemia [29,30]. Moreover, oxidative stress, which is a common characteristics of MetS, has also been associated with hematological abnormalities [28,31–33]. Hormonal changes [34], antihypertensive and antidiabetic medications [29,35–37], and the presence of atherosclerotic risk factors [38] are some additional factors proposed to cause hematological abnormalities in MetS patients.

Hematological abnormalities in MetS patients are risk factors for the development and progression of cardiovascular disease outcomes, ischemic stroke, congestive heart failure, and chronic kidney disease [29,39,40]. Metabolic disorders, including hyperglycemia and hypercholesterolemia, have been associated with hematopoietic disruption [41]. Evidence also indicates the existence of several defects in the bone marrow microenvironment in diabetes mellitus (DM) [42] as well as impaired bone marrow metabolism in IR [43]. Changes in hematological parameters brought on by IR contribute to the higher cardiovascular mortality linked to MetS [44]. Hence, hematological abnormalities in MetS patients, can further intensify the risk of morbidity and mortality, and negatively impact their quality of life [40].

Elucidating the relationship between hematological abnormalities and MetS is of great benefit for the early prevention, detection, and treatment of these abnormalities, which could reduce the morbidity and mortality associated with them and would help offer the best possible care for MetS patients. However, very few studies have been conducted to determine the magnitude and associated factors of hematological abnormalities among MetS patients in Africa, notably in Ethiopia. Particularly in the study area, no such kind of study was conducted so far. Therefore, the aim of this study was to determine the magnitude and associated factors of hematological abnormalities among MetS patients at the University of Gondar comprehensive specialized hospital, Northwest Ethiopia. The findings of this study might serve as a baseline to pave the way for other researchers to conduct further related studies. It will also be an input for policymakers, and other stakeholders to develop interventions on routine screening, and effective management of hematological abnormalities and their associated factors among MetS patients.

## Materials and methods

### Study design, period and setting

A cross-sectional study was conducted from March to May, 2022 at the chronic illness follow-up clinic of University of Gondar comprehensive specialized hospital. Gondar is a historic town and the seat of the central Gondar administrative zone in the Amhara regional state. It is situated 738 kilometers to the northwest of Addis Ababa, the capital of Ethiopia, and 180 kilometers from Bahir Dar, the regional state's capital. With an elevation of 2133 meters above sea level, the town is located at a latitude and longitude of 12˚36'N and 37˚28'E, respectively. The hospital is one of the largest teaching hospitals in the Amhara regional state, providing surgical, medical, pediatric, gynecologic, obstetric, oncologic, and ophthalmologic services to more than 7 million residents of the town and surrounding areas. The hospital has about 700

inpatient beds, intensive care units, an operating room, a fistula center, outpatient departments, and multiple wards. The chronic illness follow-up clinic has been providing services for many patients with MetS and MetS components.

## Population

All patients with MetS attending the University of Gondar comprehensive specialized hospital were considered as the source population, while patients with MetS attending the University of Gondar comprehensive specialized hospital during the study period were taken as the study population. All patients diagnosed with MetS and who fulfill the IDF criteria [4] for MetS were included in the study, however, MetS patients with known hematological conditions, chronic kidney, heart, and liver disease patients, patients with human immunodeficiency virus infection, patients with malignancy including hematological malignances, patients who were critically ill and unable to move without assistance, pregnant women, and patients who had received a blood transfusion and had a history of acute or chronic bleeding within three months of enrolment were excluded from the study.

## Variables

Anemia, leukopenia, leukocytosis, thrombocytopenia, and thrombocytosis were considered as dependent variables. On the other hand, the independent variables were: socio-demographic characteristics: age, sex, marital status, educational status, residence, and occupational status; behavioral characteristics: alcohol consumption, tobacco use, habit of physical activity; dietary characteristics; anthropometric variables: waist circumference (WC), weight, height, body mass index (BMI); clinical variables: systolic blood pressure (SBP), diastolic blood pressure (DBP), medication intake status, family history of MetS, duration of MetS, number of MetS components, and laboratory data: HDL-C, TG, fasting blood sugar (FBS) level, intestinal and malaria parasite examination result.

## Sample size and sampling technique

The sample size (n) calculation was based on a single population proportion formula, [n = (Z $\alpha/2)^2$ p (1-p)/d$^2$], considering the following assumptions: 95% confidence interval (CI) (Z$\alpha$/2 = 1.96), 5% margin of error (d) and, the expected prevalence (p) of hematological abnormalities as 50%, since there were no reports regarding the prevalence of hematological abnormalities among MetS patients in Ethiopia during the study period. Accordingly, the sample size was calculated as follows: n = (Z $\alpha/2)^2$p (1-p)/d$^2$ = (1.96)$^2$ x 0.5 (1–0.5)/ (0.05)$^2$ = 384. Therefore, a total of 384 MetS patients were included in the study. A systematic random sampling technique was applied to select study participants at a regular interval (K) of two.

## Definitions

**Metabolic syndrome** was defined by the IDF criteria [4], accordingly for a person to be defined as having MetS, they must have central obesity (defined as WC of $\geq$ 94 cm for men and $\geq$ 80 cm for women) plus any two of the four additional factors. These four factors are: raised TG level: $\geq$ 1.7 mmol/l (150 mg/dl), or specific treatment for this lipid abnormality; reduced HDL-C: < 1.03 mmol/l (40 mg/dl) in males and < 1.29 mmol/l (50 mg/dl) in females, or specific treatment for this lipid abnormality; raised BP: SBP $\geq$ 130 or DBP $\geq$ 85mmHg, or treatment of previously diagnosed hypertension; raised fasting plasma glucose: $\geq$ 5.6 mmol/ l (100 mg/dl), or previously diagnosed T2DM.

**Anemia** was defined by considering World health organization (WHO) criteria [45] and accordingly, anemia was defined as Hemoglobin (Hb) value <13g/dl for males and <12g/dl for females. Anemia severity was classified as mild (males: 11–12.9 g/dl; females: 11–11.9 g/dl), moderate (8–10.9 g/dl), and severe (<8 g/dl). Anemia was also classified by mean corpuscular volume (MCV) as microcytic (MCV< 80 fl), normocytic (MCV 80–100 fl), and macrocytic (MCV >100 fl) [46].

Hematological and immunological reference intervals for the adult population in the state of Amhara, Ethiopia [47] was used for defining the remaining hematological abnormalities and accordingly, **Leukopenia** was defined as a total white blood cell (WBC) count of less than $3 \times 10^3$ cells/$\mu$L, whereas **Leukocytosis** was defined as a total WBC count greater than $11.2 \times 10^3$ cells/$\mu$L. **Thrombocytopenia** was defined as a platelet count below $90 \times 10^3$ cells/$\mu$L, whereas **Thrombocytosis** was defined as a platelet count above $399 \times 10^3$ cells/$\mu$L.

**Tobacco use** was defined as the use of smoking or smokeless tobacco products such as manufactured cigarettes, pipes, cigars, shisha, chewing tobacco, and others. **One standard alcohol drink** was defined as a drink with a net alcohol content of approximately 10g of ethanol. For example, 1 standard bottle of regular beer (285ml), 1 single measure of spirits (30ml), and 1 medium size glass of wine (120ml). **Vigorous-intensity activities** were defined as activities that require hard physical effort and cause large increases in breathing or heart rate, like carrying or lifting heavy loads, digging, or construction work, whereas **moderate-intensity activities** were defined as activities that require moderate physical effort and cause small increases in breathing or heart rate, like brisk walking, carrying light loads, or cleaning. **Body mass index** was classified as: underweight (<18.5 kg/m$^2$), normal weight (18.5–24.9 kg/m$^2$), overweight (25–29.9 kg/m$^2$), and obese ($\geq$30 kg/m$^2$) [48].

## Data collection and laboratory methods

**Socio-demographic and lifestyle characteristics data.** Socio-demographic and lifestyle characteristics data such as age, sex, marital status, residence, alcohol consumption, tobacco use, habit of physical activity, and dietary characteristics were collected using pre-tested structured questionnaires via face-to-face-interview by trained nurses. The questionnaires were developed by using the WHO stepwise approach to non-communicable disease risk factor surveillance manual and by reviewing different literature [14,48–50].

**Anthropometric measurements and clinical characteristics data.** Anthropometric data were collected following the WHO stepwise approach to non-communicable disease risk factor surveillance manual [48]. Body weight (kg) to the nearest 0.1 kg and height (m) to the nearest 0.1 cm were measured, without shoes and in light clothing, using a portable weighting scale, which has an attached height scale. The BMI was calculated as body weight in kilograms (kg) divided by the square of height in meters (m$^2$). Waist circumference was measured at the midpoint between the lower margin of the last palpable rib and the top of the iliac crest (hip bone), at the end of a normal expiration and with the arms relaxed at the sides, using a flexible and non-stretchable tape measure.

Clinical characteristics data such as current medication intake status were collected by reviewing the patient's medical records using a checklist. Blood pressure was taken from the left arm at the heart level using a mercury sphygmomanometer and stethoscope in a sitting position after at least 15 minutes of rest. To improve the reliability, duplicate measurements (at least three minutes apart) were taken and the mean values were recorded as the final BP of the patient.

**Hematological analysis and biochemical profiles.** Complete blood count (CBC) was determined by collecting 3 milliliters of venous blood sample using a disposable syringe under

aseptic conditions by laboratory professionals. The blood was transferred to Dipotassium Eth-ylene diamine-tetraacetic acid (K2-EDTA) tube and analyzed using Beckman Coulter UniCel DxH 800 (Beckman Coulter, USA) automated five differential hematology analyzer. The ana-lyzer uses the coulter principle, spectrophotometry, and VCSn (volume, conductivity, and light scatter) technology. The manufacturer's instructions were strictly followed to perform the analysis. Biochemical profiles such as FBS, TG, and HDL-C results on the date of data collec-tion were collected by reviewing the patient's medical records using a checklist.

**Parasitological examinations.**  Pea-size (1gram) fresh stool sample was collected from each study participant in a carefully labeled, clean, dry, and leak-proof stool container. The saline wet mount technique for stool examination was done to examine stool specimens. One drop of physiological saline (0.85%) and a stool equivalent to a match stick head (2 mg) was emulsified using wooden applicator sticks on a clean and grease-free slide, then a coverslip was applied over a uniform suspension without creating bubbles. Finally, helminthes eggs, larvae, or cysts and/or trophozoans of protozoans were examined under low power objective (10x) and high power objective (40x) microscope lenses.

Febrile patients with MetS were screened for malaria parasites. Thick and thin blood films were prepared and allowed to dry in the air. The thin blood film was fixed with methanol, and both thin and thick blood films were stained with 10% Giemsa stain for 10 minutes. Following a clean water wash, the slides were air dried, and finally, the stained slides were examined microscopically by laboratory technologists. When thick blood films were reported negative after 100 fields were examined under a 100x oil immersion objective, malaria was ruled out.

## Data quality control

To ensure consistency, the questionnaires were initially prepared in English, translated into the local language (Amharic), and then back into English. Prior to the actual data collection, the questionnaire was pretested to ensure its practicability and consistency, and minor revi-sions were made. The pre-test was done on 5% of the actual sample size at Maraki health cen-ter, Gondar, Northwest Ethiopia. Training was given to data collectors about the study's purpose and significance, confidentiality, study participants' rights, consenting, interview techniques, laboratory test procedures, and quality control. Socio-demographic, anthropomet-ric, behavioral, dietary, and clinical characteristics data were collected by trained clinical nurses under the supervision of the investigator, and the laboratory tests were performed by medical laboratory technologists. Furthermore, the investigator closely followed up and fre-quently inspected the data collection process to ensure the completeness, accuracy, clarity, and consistency of the data and gave timely feedback to the data collectors.

The automated hematology analyzer's performance was evaluated by running quality con-trol reagents before running the patient's sample. The manufacturer's instructions were strictly followed for each quality control reagent, and the integrity of samples and reagents was regu-larly checked. The quality of the normal saline solution was inspected and stool specimens were checked for their quality upon reception. The quality of the Giemsa stain was evaluated using known malaria positive and negative slides. Pre-analytical, analytical, and post-analytical stages of quality assurance were strictly followed by using standard operating procedures (SOPs) of the University of Gondar comprehensive specialized hospital laboratory.

## Data analysis and interpretation

The data was cleaned, checked for completeness, and then entered into EpiData version 3.1 statistical software. Then it was exported to Stata version 14.0 (StataCorp, Texas, USA) soft-ware for analyses. Descriptive statistics were used to summarize the characteristics of the study

participants, and the results were presented in tables and figures. The Shapiro–Wilk test was used to check normality of the data and the appropriate summary measures were utilized accordingly. Bivariate and multivariate binary logistic regression models were fitted to identify factors associated with hematological abnormalities among MetS patients. Independent variables having a p-value less than 0.25 in bivariate analyses were included in the multivariate analyses to control confounders. Crude odds ratio (COR) and adjusted odds ratio (AOR) with the corresponding 95% CI was calculated to show the strength of association. A P-value of < 0.05 was considered statistically significant. Hosmer-Lemeshow goodness-of-fit test was used to check the model goodness-of-fit, and with the included independent variables, the final model was well-fitted.

## Ethical consideration

The study was conducted after it had been reviewed and approved by the Research and Ethical Review Committee of the School of Biomedical and Laboratory Sciences, College of Medicine and Health Sciences, University of Gondar (Reference number: SBMLS/182/14). A permission letter to conduct the study was obtained from the University of Gondar Comprehensive Specialized Hospital Chief Clinical Director. The objective of the research was elucidated and written informed consent was obtained from the study participants. Participation in the study was voluntary and refusal was possible. To ensure confidentiality of data, the study participants' name was omitted; instead, they were identified using their card number as well as by using unique identification codes, and unauthorized persons didn't have access to the collected data. The research participants received no payment or other form of compensation. For proper patient care and prognosis, abnormal findings were linked to physicians.

## Results

### Socio-demographic characteristics

A total of 384 MetS patients participated in the study. Of the participants, 239 (62.2%) were females. Their ages ranged from 33 to 87 years with a mean age of 60.8 ± 10.2 years. The majority 184 (47.9%) of the study participants were above the age of 60 years. Two hundred five (53.4%) respondents were married and 127 (33.1%) had no formal education. The majority 365 (95.1%) were living in urban areas and 147 (38.3%) were housewives (Table 1).

### Behavioral characteristics

The study revealed that 45 (11.7%) of the participants had used tobacco products at least once in their lives, only 9 (20%) of these being current smokers who smoke an average of 4.6±2.3 cigarettes each day. Furthermore, 283 (73.7%) of the study participants had a history of alcohol consumption at least once in their lifetime. Ninety-five participants were current alcohol users, with 51 (53.7%) of them consuming at least one standard alcoholic drink 1–3 days per month. Ninety-five (24.7%) of the participants were engaged in vigorous-intensity activity, with 39 (41.1%) of them doing so 1–3 days per week. Moreover, 267 (69.5%) of the respondents were engaged in moderate-intensity activity, with 161 (60.3%) of them doing so every day. On an average day, the respondents spent 4 hours sitting or reclining (IQR: 2, 6) (Table 2).

### Dietary characteristics

During the seven days prior to the interview, the two food groups that were consumed most frequently (daily) were cereals/grains/tubers and oil/fat/butter. Thus, 380 (99.0%) of the participants consumed cereals/grains/tubers on a daily basis, while 365 (95.1%) consumed foods

**Table 1. Socio-demographic characteristics of MetS patients at the University of Gondar comprehensive specialized hospital, Northwest Ethiopia, 2022.**

| Variable | Category | Frequency(n = 384) | Percentage (%) |
|---|---|---|---|
| Gender | Male | 145 | 37.8 |
| | Female | 239 | 62.2 |
| Age(years) | <45 | 22 | 5.7 |
| | 45–60 | 178 | 46.4 |
| | >60 | 184 | 47.9 |
| Marital status | Single | 36 | 9.4 |
| | Married | 205 | 53.4 |
| | Divorced | 49 | 12.7 |
| | Widowed | 94 | 24.5 |
| Educational status | No formal education | 127 | 33.1 |
| | Primary education | 105 | 27.3 |
| | Secondary education | 75 | 19.5 |
| | Higher education | 77 | 20.1 |
| Residence | Urban | 365 | 95.0 |
| | Rural | 19 | 5.0 |
| Occupation | Government employee | 57 | 14.8 |
| | Non-government employee | 16 | 4.2 |
| | Farmer | 12 | 3.1 |
| | Self-employed | 65 | 16.9 |
| | Housewife | 147 | 38.3 |
| | Retired | 69 | 18.0 |
| | Unemployed | 18 | 4.7 |

made from oil/fat/butter daily. However, in the seven days before the interview, 342 (89.1%), 287 (74.7%), 253 (65.9%), and 180 (46.9%) of the participants, respectively, did not consume processed foods, sugar/honey, eggs, and dairy products. Moreover, 198 (51.6%) participants did not add salt while cooking or preparing foods in their homes (Fig 1).

## Anthropometric and clinical characteristics

The median BMI of the participants was 26.3 (IQR: 24, 29), with 175 (45.6%) of the participants being overweight. Their WC ranged from 80 to 140 cm with a median of 100 cm (IQR: 96,107). Two hundred twelve (55.2%) of the participants had an SBP of $\geq$130 mmHg, with a median SBP of 130 (IQR: 120,132.5). Whereas, the mean DBP of the respondents was 78.9±8.7mmHg. The median duration of MetS among the participants was 8 years (IQR: 4, 13), ranging from 15 days to 41 years. Regarding the number of MetS components, half (48.7%) of the patients had 4 MetS components. All the patients were taking medications for the management of their MetS, of which, 348 (90.6%) and 308 (80.2%) patients were taking medications for the management of diabetes and hypertension, respectively. Intestinal parasites and malaria were found in 9 (2.3%) and 2 (0.5%) of the subjects, respectively. Out of 9 patients infected with intestinal parasites, the majority (55.6%) were infected with Ascaris lumbricoides (Table 3).

## Hematological and biochemical profile of the study participants

The altitude- and smoking-adjusted Hb level of MetS patients ranges from 4.1 to 20.1 g/dl, with a median value of 14 g/dl (IQR: 13.1, 15.1). The median WBC count was $6.5 \times 10^3$cells/µL (IQR: 5.3, 7.8) with a range of 2.8 to $15.7 \times 10^3$cells/µL. The platelet count ranges from 18 to $808 \times 10^3$cells/µL, with a median value of $223 \times 10^3$cells/µL (IQR: 185, 269.5). In terms of the bio-chemical profiles, the participants' median FBS, HDL-C, and TG levels were 143mg/dl (IQR: 114, 180), 47mg/dl (IQR: 36, 56), and 167.5mg/dl (IQR: 132, 219.5), respectively (Table 4).

**Table 2. Behavioral characteristics of MetS patients at the University of Gondar comprehensive specialized hospital, Northwest Ethiopia, 2022.**

| Variables | Categories | Frequency | Percentage |
|---|---|---|---|
| History of tobacco use | Yes | 45 | 11.7 |
| | No | 339 | 88.3 |
| Current use of tobacco products (n = 45) | Yes | 9 | 20 |
| | No | 36 | 80 |
| Amount of current cigarette smoking per day (n = 9) | <5 cigarette sticks a day | 6 | 66.7 |
| | ≥5 cigarette sticks a day | 3 | 33.3 |
| History of alcohol consumption | Yes | 283 | 73.7 |
| | No | 101 | 26.3 |
| Current use of alcohol (n = 283) | Yes | 95 | 33.6 |
| | No | 188 | 66.4 |
| Amount of standard alcoholic drink (n = 95) | 1–2 days per week | 27 | 28.4 |
| | 3–4 days per week | 4 | 4.2 |
| | 5–6 days per week | 7 | 7.4 |
| | Daily | 6 | 6.3 |
| | 1–3 days per month | 51 | 53.7 |
| Engaged in vigorous-intensity activity | Yes | 95 | 24.7 |
| | No | 289 | 75.3 |
| Frequency of vigorous-intensity activity (n = 95) | 1–3 days per week | 39 | 41.1 |
| | 4–6 days per week | 18 | 18.9 |
| | Everyday | 38 | 40.0 |
| Engaged in moderate-intensity activity | Yes | 267 | 69.5 |
| | No | 117 | 30.5 |
| Frequency of moderate-intensity activity (n = 267) | 1–3 days per week | 49 | 18.4 |
| | 4–6 days per week | 57 | 21.3 |
| | Everyday | 161 | 60.3 |
| The usual way to get to and from places | Walk | 244 | 63.5 |
| | Transport | 140 | 36.5 |
| Time spent sitting or reclining on a typical day | <4 hours a day | 166 | 43.2 |
| | 4–8 hours a day | 201 | 52.4 |
| | >8 hours a day | 17 | 4.4 |

## The magnitude of hematological abnormalities among MetS patients

**The magnitude of anemia among MetS patients.** Fifty-one of the MetS patients were found to be anemic, resulting in an overall magnitude of 13.3% (95% CI: 9.9–16.7%). Of them, 29 (56.9%) were males, 42 (82.4%) were urban residents, 26 (51.0%) were aged >60 years, 20 (39.2%) were housewives, and 24 (47.1%) had 4 MetS components. Fourty-eight (94.1%) and 47 (92.2%) of the anemic patients had DM and hypertension, respectively. Of the anemic MetS patients, 32 (62.7%), 13 (25.5%), and 6 (11.8%) were found to be mildly anemic, moderately anemic, and severely anemic, respectively. Among patients with mild anemia 30 (93.8%) had DM while all 13 patients with moderate anemia had hypertension (Fig 2). Based on their MCV, 74.5%, 21.6%, and 3.9% of the anemic patients had normocytic, microcytic, and macrocytic type of anemia, respectively (Fig 3).

**The magnitude of leukopenia, leukocytosis, thrombocytopenia and thrombocytosis among MetS patients.** The magnitude of leukopenia, leukocytosis, thrombocytopenia, and thrombocytosis was found to be 0.5% (95% CI: 0.2–1.2%), 2.9% (95% CI: 1.2–4.5%), 1.6% (95% CI: 0.3–2.8%) and 2.3% (95% CI: 0.8–3.9%), respectively (Fig 4). All patients with leukocytosis were urban residents and the majority (72.7%) were married. The prevalence of thrombocytosis was highest in patients aged 45–60 years (66.7%), while it was none among rural residents.

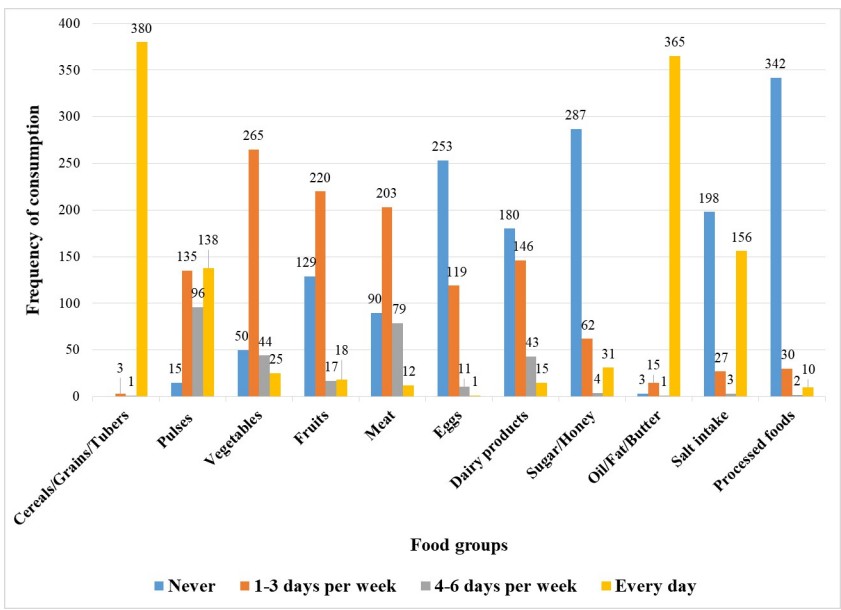

**Fig 1. Dietary characteristics of MetS patients at the University of Gondar comprehensive specialized hospital, Northwest Ethiopia, 2022.**

### Factors associated with anemia among MetS patients

Bivariate and multivariate binary logistic regression analyses were performed to identify factors associated with anemia among MetS patients. In the bivariate analysis, sex, educational status, residence, occupational status, moderate-intensity activity, vegetable consumption, sugar consumption, BMI, duration of MetS, antihypertensive medication, antidyslipidemic medication, FBS level, and TG level were significantly associated with anemia at p-value < 0.25. After adjusting potential confounders, multivariate logistic regression analysis affirmed that sex, residence, BMI, antihypertensive medication, and TG level were found to be significant predictors of anemia among MetS patients.

Male patients were 2.65 times more likely to have anemia than females (AOR = 2.65, 95% CI: 1.14, 6.20). The occurrence of anemia was 5.79 times higher among patients living in a rural areas (AOR = 5.79, 95% CI: 1.72, 19.51). Overweight or obese participants had a 68% decreased risk of acquiring anemia (AOR = 0.32, 95% CI: 0.16, 0.64) in contrast to patients with a normal weight. The odds of anemia was 3.85 times higher among patients who were taking antihypertensive medication (AOR = 3.85, 95% CI: 1.16, 12.78). Patients with elevated TG level (TG ≥150 mg/dl) were 2.21 times more likely to develop anemia (AOR = 2.21, 95% CI: 1.03, 4.75) (Table 5).

## Discussion

This study aimed to determine the magnitude and associated factors of hematological abnormalities among MetS patients. The study showed that anemia was the most prevalent hematological abnormality found among MetS patients, followed by leukocytosis and thrombocytosis. Sex, residence, BMI, antihypertensive medication, and TG level were identified as associated factors of anemia among MetS patients.

In the current study, the overall magnitude of anemia among MetS patients was 13.3% (95% CI: 9.9–16.7%). According to the WHO classification of public health significance of

**Table 3. Anthropometric and clinical characteristics of MetS patients at the University of Gondar comprehensive specialized hospital, Northwest Ethiopia, 2022.**

| Variables | Categories | Frequency | Percentage |
|---|---|---|---|
| BMI | Normal weight | 131 | 34.1 |
| | Overweight | 175 | 45.6 |
| | Obesity | 78 | 20.3 |
| SBP(mmHg) | <130 | 172 | 44.8 |
| | ≥130 | 212 | 55.2 |
| DBP(mmHg) | <85 | 296 | 77.1 |
| | ≥85 | 88 | 22.9 |
| Duration of MetS | <5 years | 112 | 29.2 |
| | 5–10 years | 154 | 40.1 |
| | >10 years | 118 | 30.7 |
| Number of MetS components | 3 MetS components | 102 | 26.6 |
| | 4 MetS components | 187 | 48.7 |
| | 5 MetS components | 95 | 24.7 |
| Antihypertensive medication | Yes | 308 | 80.2 |
| | No | 76 | 19.8 |
| Antidiabetic medication | Yes | 348 | 90.6 |
| | No | 36 | 9.4 |
| Antidyslipidemic medication | Yes | 266 | 69.3 |
| | No | 118 | 30.7 |
| Family history of MetS or its components | Yes | 173 | 45.1 |
| | No | 211 | 54.9 |
| Type of MetS components the family members have had (n = 173) | Hypertension | 107 | 61.9 |
| | DM | 125 | 72.3 |
| | Dyslipidemia | 2 | 1.2 |
| | Abdominal obesity | 24 | 13.9 |
| Malaria | Yes | 2 | 0.5 |
| | No | 382 | 99.5 |
| Intestinal parasite | Yes | 9 | 2.3 |
| | No | 375 | 97.7 |
| Type of intestinal parasite (n = 9) | *Ascaris lumbricoides* | 5 | 55.6 |
| | *Giardia lamblia* | 3 | 33.3 |
| | *Entamoeba histolytica* | 1 | 11.1 |

Abbreviations: BMI, Body Mass Index; DM, Diabetes Mellitus; DBP, Diastolic Blood Pressure; MetS, Metabolic Syndrome; mmHg, Millimeter of Mercury; SBP, Systolic Blood Pressure.

anemia in a population [45], anemia was a mild public health problem among MetS patients in the study area. The magnitude of anemia in this study was comparable with findings in India (12.3%) [51] and Kuwait (13.0%) [52]. However, this finding was lower compared to the study conducted in Worabe, Southern Ethiopia among MetS patients (25.3%) [19], studies conducted among T2DM patients in Cameroon (41.4%) [16], Iran (30.4%) [53], Malaysia (31.7% [54] and 39.4% [20]), India (45% [55] and 66% [21]), Brazil (34.2%) [56], China (22.0%) [57], Ethiopia (20.1% [58] and 34.8% [14]), and among overweight and obese patients in Iraq (26%) [18]. In contrast, this finding was relatively higher than a study conducted by Kebede et al in Ethiopia among T2DM patients (8.06%) [59]. The possible explanation for the differences could be attributed to variations in the socio-demographic, behavioral, and dietary characteristics as well as differences in clinical characteristics such as duration of MetS, and type and number of MetS components.

The study demonstrated the level of severity of anemia among MetS patients and it revealed that the majority (62.7%) of the patients had mild anemia. This finding was in agreement with previous studies in Ethiopia [58], Malaysia [20,54] and India [55]. In contrast, the study by

**Table 4. Hematological and biochemical profile of MetS patients at the University of Gondar comprehensive specialized hospital, Northwest Ethiopia, 2022.**

| Parameters | Median(IQR) | Range |
|---|---|---|
| WBC($\times 10^3$/μL) | 6.5(5.3,7.75) | 2.8–15.7 |
| RBC($10^6$/μL) | 4.8(4.5,5.1) | 2.4–6.8 |
| Hb(g/dl) | 14(13.1,15.1) | 4.1–20.1 |
| Hct (%) | 42.2(39.6,44.9) | 16.8–61.3 |
| MCV(fl) | 88.1(85.3,91.2) | 62.4–105 |
| MCH(pg) | 31.1(29.9,32.3) | 19.5–37.7 |
| MCHC(g/dl) | 35.1(34.6,35.7) | 29.1–37.9 |
| RDW (%) | 14(13.4,14.6) | 12.2–20.3 |
| PLT($\times 10^3$/μL) | 223(185,269.5) | 18–808 |
| MPV(fl) | 8.9(8.3,9.6) | 6.6–12.1 |
| Neutrophil($\times 10^3$/μL) | 3.6(2.6,4.7) | 0.8–12 |
| Lymphocyte($\times 10^3$/μL) | 1.9(1.5,2.3) | 0.2–4.8 |
| Monocyte($\times 10^3$/μL) | 0.5(0.4,0.6) | 0.1–1.3 |
| Eosinophil($\times 10^3$/μL) | 0.2(0.1,0.4) | 0–5.2 |
| Basophil($\times 10^3$/μL) | 0.1(0,0.1) | 0–0.3 |
| FBS(mg/dl) | 143(114,180) | 31–497 |
| HDL-C(mg/dl) | 47(36,56) | 16–81 |
| TG(mg/dl) | 167.5(132,219.5) | 51–699 |

Abbreviations: FBS, Fasting Blood Sugar; Hb, Hemoglobin; Hct, Hematocrit; HDL-C, High Density Lipoprotein-Cholesterol; IQR, Inter Quartile Range; MCH, Mean Corpuscular Hemoglobin; MCHC, Mean Corpuscular Hemoglobin Concentration; MCV, Mean Corpuscular Volume; MPV, Mean Platelet Volume; PLT, Platelet; RBC, Red Blood Cell; RDW, Red Blood Cell Distribution Width; TG, Triglyceride; WBC, White Blood Cell.

Timerga et al in Ethiopia reported that moderate anemia was the common type of anemia among admitted MetS patients [19]. About the morphological classification of anemia, most (74.5%) of the patients had normocytic anemia. This was in line with previous studies conducted in Ethiopia [58], China [60], Iran [53], Malaysia [20,54], and India [55]. Reversing it, a study done in Ethiopia among MetS patients [19] and in Iraq [18] among overweight and obese patients showed that microcytic anemia was the predominant type of anemia. The increased prevalence of the mild and normocytic types of anemia in the current study could be attributed to the possible occurrence of anemia of chronic disease. This type of anemia is expected to occur in clinical conditions accompanied by mild but persistent inflammation such as MetS [61,62]. The pro-inflammatory cytokines in this chronic inflammatory state, such as IL-1, IL-6, TNF, and interferon-γ, have been linked with the suppression and apoptosis of erythroid progenitor cells [29,30]. Though anemia of chronic disease is typically defined as normocytic, normochromic anemia, as the condition worsens, it can also become microcytic and hypochromic [30].

The finding of the current study showed that gender was significantly associated with anemia. Male patients were 2.65 times more likely to develop anemia than females (AOR = 2.65, 95% CI: 1.14, 6.20). This was consistent with previous studies conducted in Ethiopia [14,50,59] and USA [63]. However, the study conducted in Pakistan found that the odds of anemia was higher in females than males [64]. Given that the female participants in this study were on average 60 years old, it is possible that the decreased prevalence of anemia among them was due to the reduced blood loss caused by menopause. Additionally, men with MetS and T2DM frequently experience hypogonadotropic hypogonadism and low testosterone levels. Since testosterone stimulates erythropoiesis, decreased testosterone levels may potentially be a contributor for the increased prevalence of anemia among men [65].

The present study indicated that the occurrence of anemia was 5.79 times higher among rural residents when compared to urban residents (AOR = 5.79, 95% CI: 1.72, 19.51). This was

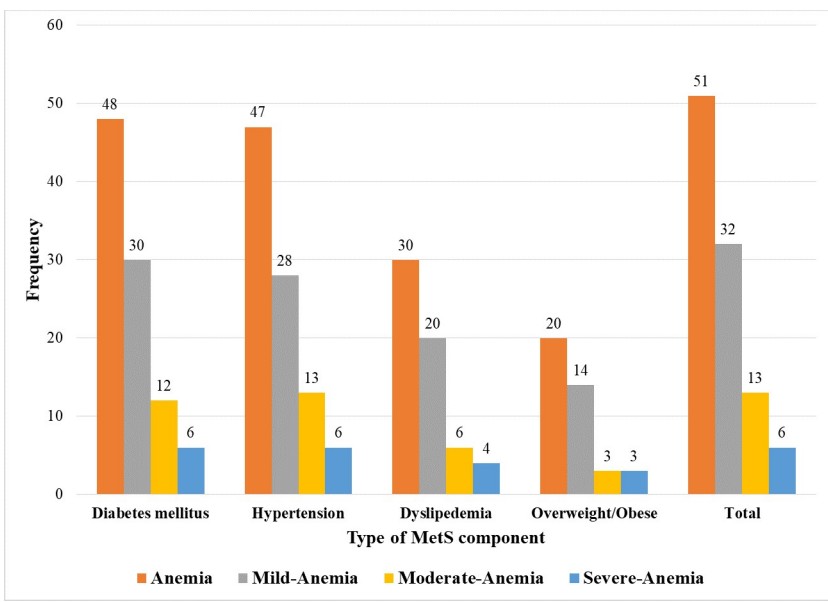

**Fig 2. Severity of anemia by MetS components among MetS patients at the University of Gondar comprehensive specialized hospital, Northwest Ethiopia, 2022.**

in line with the study conducted in Ethiopia [66]. The increased likelihood of anemia in rural inhabitants might be partly explained by dietary characteristics, considering the observed low consumption of foods such as meat, vegetables, and fruits among rural residents in the current study. This explanation seems acceptable providing that, meat is an excellent source of highly bioavailable heme iron, vegetables are a good source of non-heme iron, and fruits and vegetables contain ascorbic acid and citric acid, which aid in non-heme iron absorption [67,68].

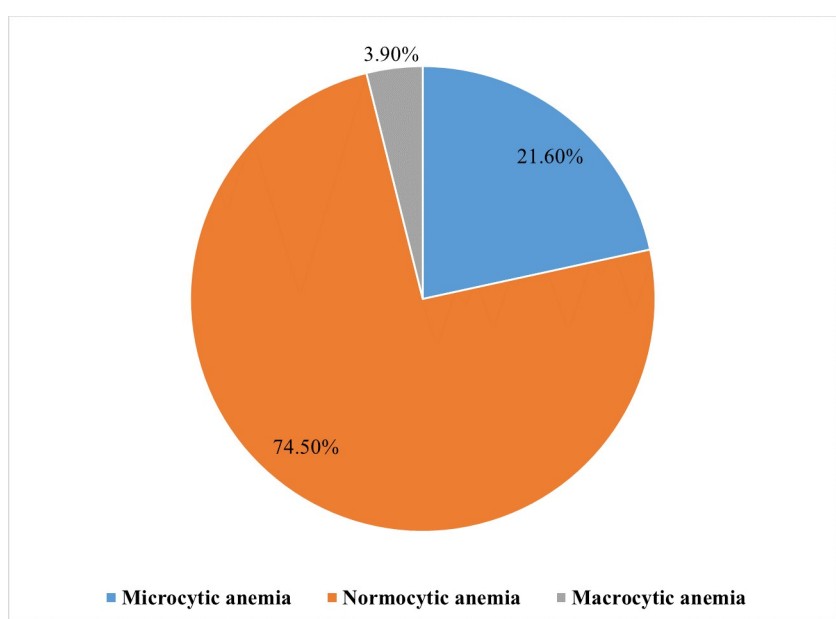

**Fig 3. Morphological classification of anemia among MetS patients at the University of Gondar comprehensive specialized hospital, Northwest Ethiopia, 2022.**

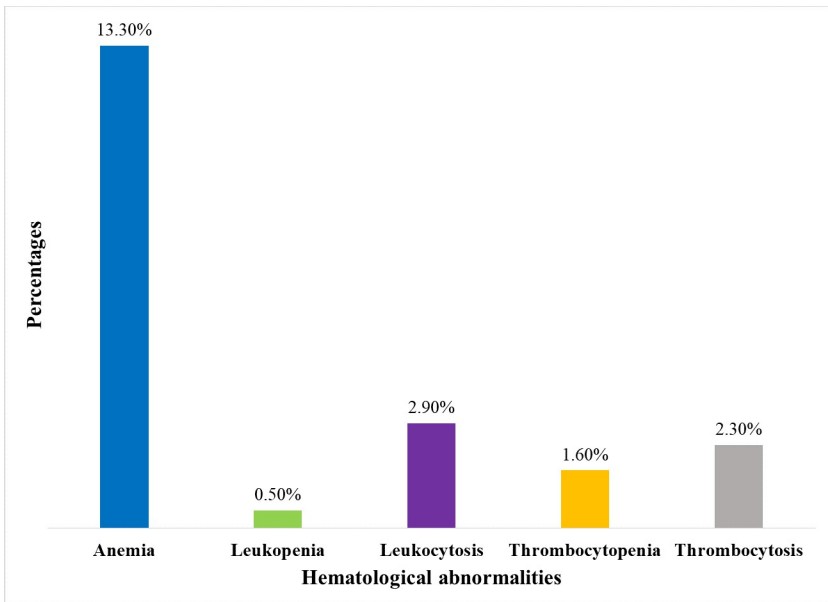

**Fig 4. Magnitude of hematological abnormalities among MetS patients at the University of Gondar comprehensive specialized hospital, Northwest Ethiopia, 2022.**

Inadequate nutrition knowledge may also be the other possible reason, considering nearly 79% of the rural residents in the present study didn't attend formal education. In addition, the inaccessibility of healthcare services in rural areas could in part contribute to the higher occurrence of anemia among rural residents [66]. However, the study conducted in Southern Ethiopia revealed an insignificant association between place of residence and anemia occurrence among MetS patients [19].

A lower likelihood of anemia was seen among overweight or obese MetS patients in the present study (AOR = 0.32, 95% CI: 0.16, 0.64) in contrast to patients with normal weight. This was in agreement with the study conducted in Malaysia [54], South Korea [61], and USA [69]. However, it was in contradiction to the studies conducted in Ethiopia [19] and Brazil [56]. These contradicting findings call for more research to investigate the association between BMI and the occurrence of anemia in MetS patients in Ethiopia. There is no clear explanation for the observed lower risk of anemia among overweight or obese MetS patients in the current study. However, the increased risk of obstructive sleep apnea in obese individuals, which can lead to persistent tissue hypoxia and elevated Hb levels, might be related to a lower risk of anemia. Furthermore, dietary elements crucial in erythropoiesis might be relatively low in people with low BMI, as compared to people with higher BMI. Therefore, adequate or overnutrition in overweight or obese individuals might be linked with a lower risk of anemia [61,69].

According to this study, MetS patients who were taking antihypertensive medications were 3.85 times more likely to develop anemia than their counterparts (AOR = 3.85, 95% CI: 1.16, 12.78). Antihypertensive medications such as angiotensin-converting enzyme inhibitors have been indicated to decrease levels of circulating angiotensin II, which leads to the suppression of erythroid precursors. This is because angiotensin II activates erythroid precursors through stimulation of an angiotensin II type 1 receptor on blast-forming units-erythroid [29,35]. Additionally, certain antihypertensive medications decrease circulating insulin-like growth factor 1, which has been linked with erythroid stimulation. Moreover, it has been indicated that these medications elevate plasma levels of the natural stem cell regulator N-acetyl-seryl-

**Table 5. Binary logistic regression of factors associated with anemia among MetS patients at the University of Gondar comprehensive specialized hospital, Northwest Ethiopia, 2022.**

| Variable | Categories | Anemia | | COR(95% CI) | P-value | AOR(95% CI) | P-value |
|---|---|---|---|---|---|---|---|
| | | Yes | No | | | | |
| Sex | Female | 22 | 217 | l* | | l* | |
| | Male | 29 | 116 | 2.47(1.36–4.49) | 0.003 | 2.65(1.14–6.20) | 0.024** |
| Educational status | No formal education | 18 | 109 | 1.95(0.74–5.16) | 0.176 | 1.90(0.52–6.90) | 0.330 |
| | Primary education | 18 | 87 | 2.45(0.92–6.50) | 0.072 | 2.12(0.68–6.66) | 0.197 |
| | Secondary education | 9 | 66 | 1.61(0.54–4.78) | 0.388 | 1.66(0.49–5.62) | 0.415 |
| | Higher education | 6 | 71 | l* | | l* | |
| Residence | Urban | 42 | 323 | l* | | l* | |
| | Rural | 9 | 10 | 6.92(2.66–18.01) | 0.000 | 5.79(1.72–19.51) | 0.005** |
| Occupation status | Gov't/Non-gov't employed | 6 | 67 | l* | | l* | |
| | Self-employed | 17 | 60 | 3.16(1.17–8.55) | 0.023 | 1.83(0.56–5.91) | 0.316 |
| | Unemployed | 28 | 206 | 1.52(0.60–3.82) | 0.376 | 1.39(0.46–4.18) | 0.557 |
| Moderate intensity activity | Yes | 31 | 236 | l* | | l* | |
| | No | 20 | 97 | 1.57(0.85–2.89) | 0.147 | 1.63(0.79–3.34) | 0.185 |
| Vegetable consumption | ≤3 days per week | 46 | 269 | 2.19(0.84–5.73) | 0.111 | 1.13(0.39–3.31) | 0.817 |
| | >3 days per week | 5 | 64 | l* | | l* | |
| Sugar consumption | ≤3 days per week | 44 | 305 | l* | | l* | |
| | >3 days per week | 7 | 28 | 1.73(0.71–4.21) | 0.224 | 1.72(0.62–4.77) | 0.296 |
| BMI | Normal weight | 31 | 100 | l* | | l* | |
| | Overweight/Obese | 20 | 233 | 0.28(0.15–0.51) | 0.000 | 0.32(0.16–0.64) | 0.001** |
| Duration of MetS | <5 year | 12 | 100 | l* | | l* | |
| | 5–10 year | 19 | 135 | 1.17(0.54–2.53) | 0.684 | 1.29(0.54–3.07) | 0.564 |
| | >10 year | 20 | 98 | 1.70(0.79–3.67) | 0.175 | 1.62(0.67–3.90) | 0.279 |
| Antihypertensive medication | Yes | 47 | 261 | 3.24(1.13–9.30) | 0.029 | 3.85(1.16–12.78) | 0.028** |
| | No | 4 | 72 | l* | | l* | |
| Antidyslipidemic medication | Yes | 30 | 236 | 0.59(0.32–1.08) | 0.085 | 0.59(0.27–1.31) | 0.197 |
| | No | 21 | 97 | l* | | l* | |
| FBS | <100 mg/dl | 11 | 44 | l* | | l* | |
| | ≥100 mg/dl | 40 | 289 | 0.55(0.26–1.16) | 0.117 | 0.74(0.31–1.72) | 0.481 |
| TG | <150 mg/dl | 16 | 140 | l* | | l* | |
| | ≥150 mg/dl | 35 | 193 | 1.59(0.84–2.98) | 0.151 | 2.21(1.03–4.75) | 0.042** |

Note: l* Reference category;

** Significant at p-value < 0.05.

Abbreviations: AOR, adjusted odds ratio; BMI, Body Mass Index; COR, crude odds ratio; FBS, Fasting Blood Sugar; Gov't, Government; MetS, Metabolic Syndrome; TG, Triglyceride.

aspartyl-lysyl-proline, which inhibits pluripotent hematopoietic stem cells recruitment and reduces red cell precursor's proliferation [35,36].

The level of TG is one of the factors significantly associated with the presence of anemia in the current study. It was observed that MetS patients with elevated TG level (TG≥150 mg/dl) were found to be 2.21 times more prone to anemia as compared to their counterparts (AOR = 2.21, 95% CI: 1.03, 4.75). This finding was consistent with the study conducted in Southern Ethiopia [19] which showed that MetS patients with dyslipidemia were more likely to be anemic as compared to patients without dyslipidemia. The observed increased likelihood of anemia among patients with elevated TG level in the current study lacks an adequate explanation. However, it has been revealed that hyperlipidemia, notably, hypertriglyceridemia, is linked with higher levels of hemolysis, since elevated lipid concentrations may modify the lipid composition of the erythrocyte membrane, and results in higher erythrocyte fragility [70].

Leukocytosis was identified in 2.9% (95% CI: 1.2–4.5%) of the participants, making it the second most prevalent hematological abnormality. It is a prevalent feature in patients with MetS, even though the percentage of leukocytosis was not mentioned in many studies. Studies conducted in China [23,71], Thailand [72], USA [73], Iran [22], Italy [74], Spain [75], and Egypt [38] reported a higher leukocyte count in MetS patients. Insulin resistance and abdominal obesity are increasingly viewed as chronic inflammatory conditions, leading to the production of a wide variety of pro-inflammatory cytokines and chemokines, including IL-6, monocyte chemoattractant protein-1, TNF-α and other inflammatory factors [76–78]. It's plausible that an active cytokine system may lead to an elevated leukocyte count since cytokines are strong inducers of leukocyte differentiation [79]. Furthermore, hypertension, hyperlipidemia, or hyperglycemia in MetS can cause endothelial dysfunction and subsequent adherence of leukocytes to the vascular wall. As a result, additional cytokines and chemokines are produced, which leads to leukocyte differentiation [26]. Finally, some hormones, such as cortisol or insulin itself, are additional potential links because these elements are known to be elevated in MetS and subsequently drive leukocyte propagation [34].

In the current study, 2.3% (95% CI: 0.8–3.9%) of MetS patients had thrombocytosis. Studies conducted in China [24], Egypt [38], Taiwan [80], and Iran [81] found that MetS patients had increased platelet counts. Several mechanisms have been proposed for this: first, the presence of atherosclerotic risk factors, such as hypertension, hyperlipidemia, and hyperglycemia activates vascular endothelial cells. As a result, the production and release of pro-inflammatory cytokines will be accelerated, leading to chronic low-grade inflammation and elevated platelet count [38,82]. Moreover, it is well known that inflammatory conditions cause an increase in TPO. This might occur as a result of the pro-inflammatory cytokine IL-6, which promotes TPO transcription in the liver and raises TPO blood levels, causing thrombocytosis [28,83]. Second, IR may potentially be a factor in thrombocytosis among MetS patients [28]. In people with IR, platelet lifespan is known to be shorter, which may be associated with platelet hyperreactivity in insulin-resistant disorders like MetS. As a result, platelets are rapidly consumed, accelerating the production of more reactive platelets [84–87]. Finally, oxidative stress, which is a common feature of MetS, may also have an additional impact on thrombocytosis in MetS patients [28,33]. Increased oxidative indicators, such as thiobarbituric acid substances and glutathione, have been linked to elevated platelet counts [88]. Additionally, diets with increased antioxidant levels have been shown to reduce platelet number, which might be attributed to oxidative stress reduction [89].

Leukopenia and thrombocytopenia were the least prevalent hematological abnormalities among MetS patients in the current study, with a prevalence of 0.5% (95% CI: 0.2–1.2%) and 1.6% (95% CI: 0.3–2.8%), respectively. There are no convincing explanations for these conditions in MetS patients, and in fact, studies reporting leukopenia and thrombocytopenia among MetS patients have been very limited. Therefore, further large-scale studies are recommended to assess the magnitude of leukopenia and thrombocytopenia among MetS patients.

It is important to consider the strengths and limitations of this study while interpreting its findings. This study was one of the first laboratory-based studies to determine the magnitude and associated factors of hematological abnormalities among MetS patients. However, it had some limitations. First, since only the saline wet mount technique was performed for stool examination, the ability to detect intestinal parasites might be reduced. Second, since the questions for some variables were answered based on recall knowledge, there is a possibility of recall bias. Third, social desirability bias may cause some of the questions, such as tobacco use and alcohol consumption, to be underestimated. Fourth, RBC morphology examination and investigations into the specific causes of anemia, such as iron, Vitamin B12, and folate level, were not performed. Finally, because of its cross-sectional nature, this study was unable to

identify any causal relationships. Despite these limitations, the study is believed to have a significant impact and considerably fill a gap in the body of knowledge.

## Conclusion and recommendation

Anemia was the most prevalent hematological abnormality identified in the present study, followed by leukocytosis and thrombocytosis. According to the current study, anemia was a mild public health problem among MetS patients in the study area. Being male, rural residency, taking antihypertensive medications, and having high TG level were associated with higher likelihood of anemia among MetS patients. Conversely, being overweight or obese was associated with a reduced risk of having anemia. Therefore, for the early detection and effective management of anemia and consequently improve patients' quality of life, routine anemia screening should be incorporated for all MetS patients, especially for those with significant associated factors. Interventions such as using antihypertensive medications with less anemic impact, and lowering high TG level can all help reduce anemia among MetS patients. Additionally, more extensive longitudinal studies with a larger sample size need to be conducted to identify the cause-effect relationships of hematological abnormalities and their associated factors among MetS patients.

## Supporting information

**S1 File. English and Amharic version of the data collection tool (Questionnaire and checklist).**
(DOCX)

**S2 File. Stata raw data file.**
(DTA)

**S3 File. STROBE checklist.**
(DOCX)

## Acknowledgments

A very special thanks to the management, clinical nurses, medical laboratory technologists, and physicians of the University of Gondar comprehensive specialized hospital for their valuable support during data collection. We would like to thank the study participants for their willingness to participate and for providing the necessary information for this study.

## Author Contributions

**Conceptualization:** Amanuel Kelem, Elias Shiferaw, Tiruneh Adane.

**Data curation:** Amanuel Kelem.

**Formal analysis:** Amanuel Kelem, Elias Shiferaw, Tiruneh Adane.

**Funding acquisition:** Amanuel Kelem.

**Investigation:** Amanuel Kelem.

**Methodology:** Amanuel Kelem, Elias Shiferaw, Tiruneh Adane.

**Project administration:** Amanuel Kelem, Elias Shiferaw, Tiruneh Adane.

**Resources:** Amanuel Kelem.

**Software:** Amanuel Kelem.

**Supervision:** Amanuel Kelem, Elias Shiferaw, Tiruneh Adane.

**Validation:** Amanuel Kelem, Elias Shiferaw, Tiruneh Adane.

**Visualization:** Amanuel Kelem, Elias Shiferaw, Tiruneh Adane.

**Writing – original draft:** Amanuel Kelem.

**Writing – review & editing:** Amanuel Kelem, Elias Shiferaw, Tiruneh Adane.

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
