## [Decision Letter · Decision Letter 0]

3 Apr 2023

PONE-D-22-30096Hematological abnormalities and associated factors among metabolic syndrome patients at the University of Gondar comprehensive specialized hospital, Northwest EthiopiaPLOS ONE

Dear Dr. Bekele,

Thank you for submitting your manuscript to PLOS ONE. After careful consideration, we feel that it has merit but does not fully meet PLOS ONE’s publication criteria as it currently stands. Therefore, we invite you to submit a revised version of the manuscript that addresses the points raised during the review process.

We look forward to receiving your revised manuscript.

Kind regards,

Desalegn Admassu Ayana, Ph.D

Academic Editor

PLOS ONE

Journal Requirements:

"We are extremely grateful to the University of Gondar for providing the financial means to complete this study.."

"The study was funded by the University of Gondar.

**Additional Editor Comments:**

The comments given by the first reviewer should be addressed focusing reviewing the title (focusing on anemia) and review of the methodological issues to address the sampling and the instrument data quality issue,

Reviewers' comments:

Reviewer's Responses to Questions

**Comments to the Author**

1. Is the manuscript technically sound, and do the data support the conclusions?

Reviewer #1: Partly

Reviewer #2: Yes

2. Has the statistical analysis been performed appropriately and rigorously? 

Reviewer #1: I Don't Know

Reviewer #2: Yes

3. Have the authors made all data underlying the findings in their manuscript fully available?

Reviewer #1: Yes

Reviewer #2: Yes

4. Is the manuscript presented in an intelligible fashion and written in standard English?

Reviewer #1: Yes

Reviewer #2: Yes

5. Review Comments to the Author

Reviewer #1: Comments on: Hematological abnormalities and associated factors among metabolic syndrome patients at the University of Gondar comprehensive specialized hospital, Northwest Ethiopia

Line 1. Title page it is good to add common hematological abnormalities ….as the authors investigated limited hematology related measurements among many

Abstract section: Methods part: Authors did not mentioned who are patients of metabolic syndrome ? how they are diagnosed .

Introduction part Line 124, it is better to limit common hematological abnormalities rather than all types.

Methods part : Line 157 : authors did not indicate total number of MetS patients, please indicate that

Second sampling methods says systematic random sampling : It is good to explain the selection processes with actual numbers !!

Line 246 : Authors did not mention the time when was FBS, TG, and HDL-C testes were done if there are time variation it could have impact on the interpretation of the results unless it is stated clearly including the limitation of secondary data . Moreover try to consider method variation, do all samples analyzed with similar instruments / methods?

Line 276-277 : authors stated that “The automated hematology analyzer's performance was evaluated by running quality control reagents” I think here we need to check with known quality control materials it is not mentioned in your QA work.

On table 3 , please describe the stage of the parasites for Girdia lamblia and Entamoeba histolytica/diaper

Line 363-364 : authors mentioned altitudes and smoking adjustment Hemoglobin value is described but how the adjustment was made is not mentioned in the methods section. It is good to state that to replicate the work.

Try to avoid repeating results in text, tables and figures

The discussion looks a bit longer try to reduce by avoiding redundant information

Under limitation part it is wise to mention the biochemical parameters are taken as a secondary data not done in parallel with the hematological parameters it could have impact on the findings. Moreover, try to indicate anemia and other hematological abnormalities are happened before Mets or during Mets follow up ?

Reviewer #2: The manuscript addresses one of the important public health issues, the prevalence of undiagnosed hematological disorders among MetS study participants. The findings will give additional input to improve the quality of life among MetS-developed individuals. Also encouraged other researchers to conduct similar studies to consolidate the findings in Ethiopia, and /or worldwide. Thus, if the manuscript is published, it will have a good response to advance the medical science world.

6. PLOS authors have the option to publish the peer review history of their article (what does this mean?). If published, this will include your full peer review and any attached files.

Reviewer #1: No

Reviewer #2: **Yes: **Dr. Mistire Wolde, PhD; Associate Professor

---

## [Author Response · Author response to Decision Letter 0]

13 Apr 2023

Date: April 13,2023

To: Plos One Journal

Subject: Sending a rebuttal letter

Dear Sir/madam, Greetings!

These is a rebuttal letter that responds to each point raised by the academic editor and reviewer(s). 

For the Academic Editor

1. We ensure that our manuscript meets PLOS ONE's style requirements, including those for file naming.

2. We ensure that we removed any funding-related text from the manuscript and we agreed with the current Funding Statement which reads as follows: "The study was funded by the University of Gondar. The funders had no role in study design, data collection and analysis, decision to publish, or preparation of the manuscript." Though we have no update on our current funding statement, we included the above statement within our cover letter.

3. We included captions for our Supporting Information files at the end of our manuscript, and we updated any in-text citations to match accordingly.

4. We reviewed our reference list and we ensured that it is complete and correct. 

For Reviewer 1

• You have recommended to add other common hematological abnormalities ….but as we are in resource limited country Ethiopia, we had financial, reagent as well as equipment constraints to add other hematology related measurements. But in the future if we got this supports we may certainly measure other hematology related parameters among metabolic syndrome patients, and we just put recommendations forward to other researchers to conduct more extensive studies.

• Abstract section: we try to mention who are patients of metabolic syndrome and how they are diagnosed, but the word limit for the abstract will exceed more than 300 words and this will be against the submission guidelines of the journal.

• Introduction part Line 124, we believe that including many types of hematological abnormalities will make the study more sound, rather than including only the common ones. 

• Sampling methods says systematic random sampling: It is good to explain the selection processes with actual numbers !!

Answer: we explain the selection process with actual numbers, which can be read as follows: “A systematic random sampling technique was applied to select study participants at a regular interval (K) of two”. 

• Line 246 : Authors did not mention the time when was FBS, TG, and HDL-C testes were done if there are time variation it could have impact on the interpretation of the results unless it is stated clearly including the limitation of secondary data . Moreover try to consider method variation, do all samples analyzed with similar instruments / methods?

Answer: it says that “Biochemical profiles such as FBS, TG, and HDL-C results on the date of data collection”.so these parameters were collected on the date of the data collection. For every answer: Mets patients these tests are inevitable tests when they come for checkup and we just collect these updated and timely results on the date of the data collection. Regarding the method variation, all biochemical tests were done with the same chemistry machine and since we only take the data from the patients’ charts we believe that we don’t need to mention the way these biochemical profiles were done.

• Line 276-277 : authors stated that “The automated hematology analyzer's performance was evaluated by running quality control reagents” I think here we need to check with known quality control materials it is not mentioned in your QA work.

Answer: It is obvious that quality control reagents are known quality control materials, and we mentioned that “The automated hematology analyzer's performance was evaluated by running quality control reagents (known quality control materials)”

• On table 3, please describe the stage of the parasites for Girdia lamblia and Entamoeba histolytica/diaper

All were trophozoite stages of e.histolytica and g.lamblia 

• Line 363-364 : authors mentioned altitudes and smoking adjustment Hemoglobin value is described but how the adjustment was made is not mentioned in the methods section. It is good to state that to replicate the work.

Answer: We just put these on references as well as on the supporting informations.

Try to indicate anemia and other hematological abnormalities are happened before Mets or during Mets follow up?

Answer: Because of its cross-sectional nature, this study was unable to identify any causal relationships, and unable to indicate anemia and other hematological abnormalities are happened before Mets or during Mets follow up.

Sincerely!

Amanuel Kelem: Department of Medical Laboratory Sciences, Asrat Woldeyes Health Science Campus, Debre Berhan University, Debre Berhan, Ethiopia. E-mail: amanuelkelem@gmail.com. Tel.: +251-9-11712112

Elias Shiferaw: Department of Hematology and Immunohematology, School of Biomedical and Laboratory Sciences, College of Medicine and Health Sciences, University of Gondar, Gondar, Ethiopia. E-mail: eliasshiferaw2008@gmail.com. Tel.: +251-9-27607075

Tiruneh Adane: Department of Hematology and Immunohematology, School of Biomedical and Laboratory Sciences, College of Medicine and Health Sciences, University of Gondar, Gondar, Ethiopia. E-mail: tirunehadane01@gmail.com. Tel.: +251-9-18754325

---

## [Decision Letter · Decision Letter 1]

10 May 2023

Hematological abnormalities and associated factors among metabolic syndrome patients at the University of Gondar comprehensive specialized hospital, Northwest Ethiopia

PONE-D-22-30096R1

Dear Mr. Bekele,

We’re pleased to inform you that your manuscript has been judged scientifically suitable for publication and will be formally accepted for publication once it meets all outstanding technical requirements.

Kind regards,

Desalegn Admassu Ayana, Ph.D

Academic Editor

PLOS ONE

Additional Editor Comments (optional):

Reviewers' comments:

Reviewer's Responses to Questions

**Comments to the Author**

1. If the authors have adequately addressed your comments raised in a previous round of review and you feel that this manuscript is now acceptable for publication, you may indicate that here to bypass the “Comments to the Author” section, enter your conflict of interest statement in the “Confidential to Editor” section, and submit your "Accept" recommendation.

Reviewer #1: All comments have been addressed

2. Is the manuscript technically sound, and do the data support the conclusions?

Reviewer #1: Yes

3. Has the statistical analysis been performed appropriately and rigorously? 

Reviewer #1: Yes

4. Have the authors made all data underlying the findings in their manuscript fully available?

Reviewer #1: Yes

5. Is the manuscript presented in an intelligible fashion and written in standard English?

Reviewer #1: Yes

6. Review Comments to the Author

Reviewer #1: Authors addressed almost all comments. However, the explanation they mentioned for quality control materials and quality control reagents has to be clarified. In hematological measurements, you need to run QC materials low, high and normal to check the instrument is working. You need to mention weather you used such QC materials in your lab analysis. Similarly, authors collected TG, COL, FBS, and other data which is secondary not a primary data and nothing is mentioned the time of testing. Which means that some patients may had new data ( tested with in one week of data collection others could be months back , even year/s this could influence your interpretation. You may mention under limitation part too.

7. PLOS authors have the option to publish the peer review history of their article (what does this mean?). If published, this will include your full peer review and any attached files.

Reviewer #1: No

---

## [Editor Report · Acceptance letter]

17 May 2023

PONE-D-22-30096R1 

Hematological abnormalities and associated factors among metabolic syndrome patients at the University of Gondar comprehensive specialized hospital, Northwest Ethiopia 

Dear Dr. Bekele:

I'm pleased to inform you that your manuscript has been deemed suitable for publication in PLOS ONE. Congratulations! Your manuscript is now with our production department. 

Kind regards, 

on behalf of

Dr. Desalegn Admassu Ayana 

Academic Editor

PLOS ONE